# Social and structural factors associated with substance use within the support network of adults living in precarious housing in a socially marginalized neighborhood of Vancouver, Canada

Verena Knerich[1,2], Andrea A. Jones[2], Sam Seyedin[2], Christopher Siu[2], Louie Dinh[3], Sara Mostafavi[4,5], Alasdair M. Barr[6], William J. Panenka[2], Allen E. Thornton[7], William G. Honer[2], Alexander R. Rutherford[8]*

1 Departments of Computer Science, and Cultural Anthropology, Ludwig-Maximilians University, Munich, Germany, 2 Department of Psychiatry, University of British Columbia, Vancouver, BC, Canada, 3 Department of Computer Science, University of British Columbia, Vancouver, BC, Canada, 4 Department of Statistics, University of British Columbia, Vancouver, BC, Canada, 5 Medical Genetics, Department Office, University of British Columbia, Vancouver, BC, Canada, 6 Department of Anesthesia, Pharmacology and Therapeutics, University of British Columbia, Vancouver, BC, Canada, 7 Department of Psychology, Simon Fraser University, Burnaby, BC, Canada, 8 Department of Mathematics, Simon Fraser University, Burnaby, BC, Canada

* arruther@sfu.ca

**Data Availability Statement:** Data cannot be made publicly available due to possible privacy breaches

## Abstract

### Background

The structure of a social network as well as peer behaviours are thought to affect personal substance use. Where substance use may create health risks, understanding the contribution of social networks to substance use may be valuable for the design and implementation of harm reduction or other interventions. We examined the social support network of people living in precarious housing in a socially marginalized neighborhood of Vancouver, and analysed associations between social network structure, personal substance use, and supporters' substance use.

### Methods

An ongoing, longitudinal study recruited 246 participants from four single room occupancy hotels, with 201 providing social network information aligned with a 6-month observation period. Use of tobacco, alcohol, cannabis, cocaine (crack and powder), methamphetamine, and heroin was recorded at monthly visits. Ego- and graph-level measures were calculated; the dispersion and prevalence of substances in the network was described. Logistic mixed effects models were used to estimate the association between ego substance use and peer substance use. Permutation analysis was done to test for randomness of substance use dispersion on the social network.

and other ethical and legal obligations to the study participants. These restrictions are outlined by the University of British Columbia's Clinical Research Ethics Board and Simon Fraser University's Research Ethics Board. Inquiries regarding data can be made to the Clinical Research Ethics Board of the University of British Columbia (ethics. research.ubc.ca) and the Hotel Study principal investigator at william.honer@ubc.ca.

**Funding:** VK and ARR were supported in part by the Simon Fraser University Big Data Initiative during this research. The study was funded by Canadian Institutes of Health Research grants CBG-101827 and MOP-137103 (http://www.cihr-irsc.gc.ca/). Additional support was provided by British Columbia Mental Health and Addictions Services. WGH was supported by the Jack Bell Chair in Schizophrenia at the University of British Columbia. The funders had no role in study design, data collection and analysis, decision to publish, or preparation of the manuscript.

**Competing interests:** I have read the journal's policy and the authors of this manuscript have the following competing interests: AMB has received consulting fees or sat on advisory boards for Bristol- Myers Squibb, Eli Lilly, and Roche. WJP sat on paid advisory boards for Vitality Biopharma, Medipure Pharmaceuticals, and Vinergy Resources; has sat on the board of directors of Abbatis Bioceuticals; and is owner of Translational Life Sciences. WGH received consulting fees or sat on paid advisory boards for: the Canadian Agency for Drugs and Technology in Health, AlphaSights, Guidepoint, In Silico, Translational Life Sciences, Otsuka, Lundbeck, and Newron. This does not alter our adherence to PLOS ONE policies on sharing data and materials.

## Results

The network topology corresponded to residence (Hotel) with two clusters differing in demographic characteristics (Cluster 1 –Hotel A: 94% of members, Cluster 2 –Hotel B: 95% of members). Dispersion of substance use across the network demonstrated differences according to network topology and specific substance. Methamphetamine use (overall 12%) was almost entirely limited to Cluster 1, and absent from Cluster 2. Different patterns were observed for other substances. Overall, ego substance use did not differ over the six-month period of observation. Ego heroin, cannabis, or crack cocaine use was associated with alter use of the same substances. Ego methamphetamine, powder cocaine, or alcohol use was not associated with alter use, with the exception for methamphetamine in a densely using part of the network. For alters using multiple substances, cannabis use was associated with lower ego heroin use, and lower ego crack cocaine use. Permutation analysis also provided evidence that dispersion of substance use, and the association between ego and alter use was not random for all substances.

## Conclusions

In a socially marginalized neighborhood, social network topology was strongly influenced by residence, and in turn was associated with type(s) of substance use. Associations between personal use and supporter's use of a substance differed across substances. These complex associations may merit consideration in the design of interventions to reduce risk and harms associated with substance use in people living in precarious housing.

## Introduction

Substance use disorders, along with mental and physical illness form the "tri-morbidity" affecting socially marginalized people living in precarious housing or homelessness [1,2]. In an impoverished neighborhood of Vancouver, Canada, multiple substances including tobacco, alcohol, cannabis, stimulants (cocaine and methamphetamine) and opioids are easily accessed and commonly used [3,4]. In addition to creating substance use disorders and risk of overdose, psychosis and increased risk of exposure to viral infection are comorbidities associated with premature mortality [3–6]. Interventions focused on individuals have made impacts on overdose mortality [7], yet poor health remains a problem for many in the community. The complexity of potentially additive or interactive risk factors may create considerable heterogeneity within community members, creating challenges for interventions to improve health. A gap in knowledge of how social network structure influences substance use in the community may have important implications for developing more effective service delivery.

Personal behaviors including substance use are affected by social contacts [8–14]. Peer influence has been shown to play an important role in both the initiation and nature of substance use by youth and young adults [11,15–17]. Recent studies demonstrate that youth who are homeless or unstably housed form clustered social networks with a strong correlation between substance use of peers [15,18]. Individuals centrally located in large clusters also often have higher levels of substance use or other high risk behavior [18]. Similar results were also found for marginalized adults in inner-city neighborhoods [19] and women receiving substance use treatment [20].

It has been observed in a number of studies that spatial proximity of housing could play an important role in the formation of social ties and social networks may be best examined at the community level [10,21–23]. In a study of the role of social networks by urban youth, the spatial proximity of homes and locations for socializing was found to be important [24]. An association between social networks and locations for socializing was also found in a study of drug and alcohol use by African American men who have sex with men [25]. The influence of neighborhoods in defining social environments has been liked to risk behaviors for HIV, such as injection drug use [26].

Social ties, the structure or topology of social networks, and the dynamics of behavioral change or propagation within a network are features that may inform future intervention efforts [10,11,14,27–32]. It has been proposed that treatment outcomes could be improved by helping patients develop social networks which do not include, or at least minimize the number of substance users [33]. The importance of understanding social networks in the context of 12-step programs for young adults is highlighted in [34].

A community-based study of substance use, mental and physical health in Vancouver provides an opportunity to learn how these factors are associated [3,4]. The study was designed similarly to the Memory and Aging Project, a community-based longitudinal study that recruited elderly people from retirement communities in Chicago, and identified social network contributions to cognitive reserve [35,36]. The Hotel Study recruits participants from single room occupancy hotels in an impoverished neighborhood in Vancouver. Similar to the Memory and Aging Project, a comprehensive baseline and similar annual assessments are carried out. The Hotel Study also includes assessments allowing social networks to be mapped, and overlaid with monthly assessments of substance use. For the present analyses from this longitudinal study, we investigated how the underlying social network structure contributed to the distribution of substance use in the network, and we sought to understand whether personal substance use was associated with the substance use behaviours of social connections.

## Methods

### Participants

In Canada, precarious housing includes single room occupancy hotels (origin of the "Hotel Study" name) if the accommodation is below Canadian standards for adequacy (need for repairs), affordability (rental costs <30% of before-tax income), or suitability (makeup of bedrooms and household) [37]. The Hotel Study is an ongoing longitudinal study of mental and physical health, and substance use of adults living in precarious housing in an impoverished neighbourhood of Vancouver, Canada [38]. Community-based sampling was utilized to approach all residents of four single room occupancy (SRO) hotels located within 0.6 km$^2$.

### Ethics statement

Age 18 years or older, ability to communicate in English and provide written informed consent were required. Consent was reaffirmed at each follow up visit. The study was approved by the Research Ethics Boards of the University of British Columbia and Simon Fraser University.

### Demographic, social, and clinical descriptive assessments

Demographic information was collected at study entry. Information included the Arizona Social Support Interview Schedule, a name-generating questionnaire [39,40]. Participants were asked to report the names of all people they could either turn to for social support, or had a negative encounter with in the past month. Social support included intimate interaction,

material aid, advice, positive feedback and physical assistance [40]. It was possible to report both a positive and a negative interaction with an individual.

At baseline, and at each monthly visit thereafter, self-reported use of methamphetamine, heroin, powder cocaine, crack cocaine, cannabis, alcohol and tobacco were obtained. Self-reported substance use in this sample shows a substantial rate of agreement with urine drug screen data (methamphetamine Cohen's kappa = 0.66, cannabis kappa = 0.66, cocaine kappa = 0.67, opiates kappa = 0.70) as reported previously [41,42]. Substance use was reported for each of the four weeks prior to the monthly interview and analysed as a binary variable defined as any substance use within the past month. The interval between study visits was four weeks. For the six months following network fixation, we aligned the interview dates with the calendar month for standardisation.

## Network construction

We constructed a sociogram of social support relations between participants using the Arizona Social Support Interview Schedule data. The term ego refers to an individual in the network, connected to one or more alters through support relationships. For each alter named, additional demographic information was gathered. The numbers of alters was not limited, but to be included in the network analysis, egos and alters were required to be Hotel Study participants. Ties to network members within the study were confirmed using four sources of information. First, alter names were compared against names of study participants using a name matching algorithm. If the names matched, demographic information (age, sex, ethnicity, residence address) of study participants was also compared by the algorithm. Third, these linkages were verified against the longitudinal relational data if the alter was named on multiple assessments over time. Lastly, study staff with greater than seven years-experience with the community reviewed and verified all proposed linkages. In unresolved cases, the tie was not included in the network. This approach resembles techniques applied to the study of disease transmission and substance use in social networks [43,44]. A sociogram was constructed based on these inferred ties between study participants (n = 201) in R using the igraph package [45]. A cluster is a group of densely connected nodes. Egos without supporters within the study or not named as supporters are called isolates (n = 83). In our context, positive edges are defined as a supporting relationship with a study participant, while negative edges refer to disagreement. Outgoing ties point towards supporters or negative contacts of the ego. Incoming ties to an ego signify that the ego was named as supporter or negative contact by one of his or her alters. Ties could be reciprocated if reported by both ego and alter.

## Network measures

Gender disparity was calculated as the squared difference between gender of an ego and the gender of all alters divided by the number of supporters of the ego [0; 1]. Larger values are therefore indicative of more gender disparity between supporters. Age disparity was calculated analogously yet the difference was not squared. Therefore, negative values are indicative of an ego being younger than linked alters and positive values are indicative of an ego being older. For further details see S1 Appendix.

At the network node-level, several ego-level measures were estimated from the sociogram, including degree, betweenness centrality and assortativity. Degree is the number of alters for an individual ego, and is reported according to the directionality of the tie (in, out, total). Indegree denotes the number of ingoing ties per ego or how often ego was named as supporter by the alters. Outdegree refers to the number of outgoing edges per ego or how many supporters the ego named. Total degree is the sum of indegree and outdegree. Betweenness centrality was

calculated as the total number of shortest paths passing through a vertex. A shortest path is the minimum number of edges traversed to connect a pair of vertices. Assortativity measures the propensity of nodes to connect to similar others by certain features. Positive values for the degree assortativity denote a high propensity for connecting to nodes with a similar degree [46]. In this case, positive values indicate that high degree nodes show a preference for connecting to other high degree nodes. Gender assortativity calculates propensity of vertices connecting to alters of the same gender. It is based on the fraction of edges that connect dissimilar vertices in relation to edges that connect similar ones [46].

At the network graph-level, network density is calculated as the ratio of the number of edges and the number of possible edges within the network, allowing for loops. Lastly, the true size of ego's support network, consisting of all supporters (or negative contacts) named inside and outside the study, was estimated and referred to as alters available.

### Statistical analysis

We inspected the network diagram using the Fruchterman-Rheingold layout [47]. Hotel affiliation, gender, age and type of tie were included as the most relevant attributes. We analysed the full network including the isolates (n = 201), cluster 1 (n = 37), cluster 2 (n = 20), the other small components (n = 61), and in some cases the isolates (n = 83). For each ego, we counted the number of alters using a substance, and divided by the total number of alters for that ego (outdegree). The mean of these values for all egos in the full network, or for all egos in a part of the network (cluster, or small components) was then calculated for each of the seven substances. The same calculation was then repeated, but with the numerator limited to alters using the same substance as the ego.

All seven substances were mapped to the network (excluding the isolates) for six months and their dispersion was inspected visually. The patterns of substance use dispersion were compared with the descriptive results and verified using mixed effects models. We constructed separate logistic mixed effects models with random intercepts for each substance to estimate the association between alter substance use as the independent variable and ego substance use as the binary dependent variable over six months [48]. Time was included as a covariate, because the data on substance use are longitudinal. Additive models emerged as the best fit, where the total variance is explained by the sum of the covariates [49]. Additional mixed effects models were utilized to estimate these relationships in a subset of individuals injecting each substance to determine if route of administration was relevant. Combinations of different substance use were also considered. These approaches are further described in S2 Appendix. We used R lme4 package for the analyses [50].

To determine whether the dispersion of substance use and the association between ego and alter use was random, we conducted permutation analyses. After randomly reshuffling substance use 10,000 times on the initial network, we recalculated alter substance use and compared them with the observed values (see S3 Appendix).

## Results

### Demographic and descriptive network characteristics

Staggered recruitment for the core Hotel Study began in November 2008. By the end of September 2010, 246 of approximately 341 potential participants living in the four hotel sites (72%) joined the study. The time window from November 1, 2009 until September 30, 2010 was used to establish the network; 201 of 246 participants (82%) provided network information and were included in subsequent analyses. Networks were considered fixed for the next six months of outcome data collection.

Demographic and network characteristics are given for the full network (n = 201), Cluster 1 (n = 37), Cluster 2 (n = 20) and the small components (n = 61) in Table 1. Comparison with the demographics of other Canadian studies of homeless and precariously housed samples appears in S1 Table. Gender, age, hotel affiliation, type (positive/negative) and directionality of ties between participants are displayed as a graph excluding the isolates in Fig 1. Participants mainly named residents of their own hotel as supporters or negative connections. Accordingly, Cluster 1 corresponded 94% to Hotel A, and Cluster 2 corresponded 95% to Hotel B.

Participants from Clusters 1 and 2 differed in demographic attributes. Participants in Cluster 1 were a mean of 10.4 years younger than members of Cluster 2. There were 11% more men in Cluster 1 than in Cluster 2, and the former was more ethnically diverse, with residents having lived on average 30.5 fewer months in their residence. Participants in Cluster 1 were prone to connect with supporters of the same gender (gender disparity mean: 0.24, SD: 0.36) even though there were comparatively few women (24% vs. 35% Cluster 2). In general, women seemed to form chains of women supporters. Network characteristics also differed by cluster. Participants in Cluster 1 were densely connected within two subclusters, reported negative interactions frequently and often reciprocated support. Participants in Cluster 2 and the small components had most access to support outside of other study participants (alters available Cluster 2: mean 4.9; small components: 4.2).

## Dispersion of ego substance use on the network

Ego substance use, all substance-using alters regardless of ego use, and all substance-using alters for an ego using the same substance in the first month are displayed in Fig 2, using the same layout as Fig 1 and a color-coded representation of substance use in the first month.

**Table 1. Demographics and network characteristics of four different graph components.**

|  | Full network (n = 201) mean (SD) | Cluster 1 (n = 37) mean (SD) | Cluster 2 (n = 20) mean (SD) | Small components (n = 61) mean (SD) | Isolates (n = 83) mean (SD) |
|---|---|---|---|---|---|
| Age | 44.0 (9.4) | 37.3 (7.8) | 47.7 (6.9) | 46.9 (9.6) | 43.9 (9.1) |
| Male / female / trans (%) | 75 / 25 / 1 | 76 / 24 / 0 | 65 / 35 / 0 | 67 / 33 / 0 | 82 / 17 / 1 |
| Ethnicity: White / Indigenous / Other (%) | 55 / 32 / 12 | 51 / 27 / 21 | 60 / 40 / 0 | 61 / 30 / 10 | 51 / 35 / 14 |
| Education (grade completed) | 10.0 (2.0) | 10.2 (1.5) | 10.2 (2.3) | 9.5 (2.2) | 10.2 (2.1) |
| Time in hotel (months) | 47.1 (44.7) | 53.7 (30.6) | 84.2 (54.2) | 45.9 (45.1) | 36.0 (42.5) |
| Hotel: A / B / C / D / Other (%) | 34 / 33 / 15 / 11 / 6 | 84 / 0 / 0 / 0 / 16 | 0 / 95 / 0 /0 / 5 | 13 / 30 / 28 / 20 / 10 | 35 / 35 / 17 / 12 / 1 |
| **Network descriptive measures** |  |  |  |  |  |
| Age disparity | 0.01 (4.75) | 0.04 (4.75) | 0.21 (5.36) | -0.06 (7.24) |  |
| Gender disparity | 0.19 (0.37) | 0.24 (0.36) | 0.32 (0.44) | 0.37 (0.47) |  |
| Indegree | 0.78 (1.06) | 1.97 (1.42) | 1.25 (0.91) | 0.95 (0.67) |  |
| Outdegree | 0.78 (1.33) | 1.97 (2.19) | 1.25 (1.33) | 0.95 (0.78) |  |
| Total degree | 1.55 (2.13) | 3.95 (3.12) | 2.50 (1.61) | 1.90 (1.06) |  |
| Positive edges per ego | 0.69 (1.18) | 1.65 (1.99) | 1.20 (1.24) | 0.87 (0.64) |  |
| Negative edges per ego | 0.14 (0.49) | 0.38 (0.83) | 0.05 (0.22) | 0.23 (0.56) |  |
| Reciprocated edges per ego | 0.36 (0.64) | 0.81 (1.00) | 0.50 (0.61) | 0.52 (0.57) |  |
| Betweenness | 3.01 (10.30) | 14.14 (20.48) | 1.90 (3.04) | 0.70 (1.99) |  |
| Assortativity degree | 0.36 | 0.14 | -0.2 | 0.1 |  |
| Assortativity gender | 0.07 | 0.08 | 0.14 | 0.01 |  |
| Density | 0.01 | 0.05 | 0.06 | 0.02 |  |
| Alters available | 3.82 (2.70) | 3.22 (2.70) | 4.90 (2.69) | 4.16 (2.67) |  |

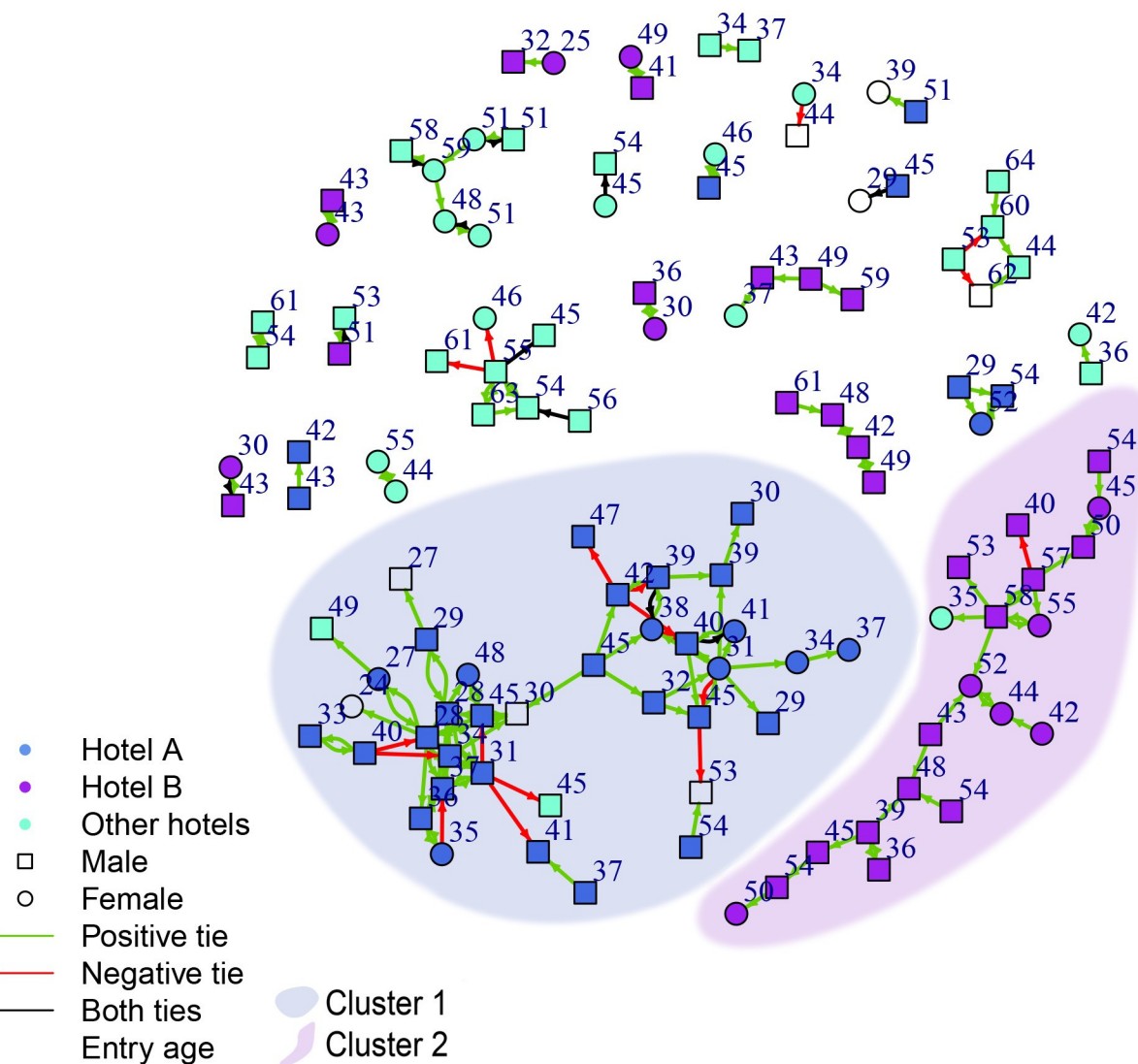

**Fig 1. Fruchterman-Rheingold layout of network diagram of SRO tenants (n = 118).** Ties point towards supporters or negative contacts as perceived by ego. Tie colour indicates relation type (positive, negative, reciprocal). Node colour indicates hotel residence and node shape indicates gender.

Dispersion across the network differed between substances (Table 2, results for injection substance use in S2 Table). Methamphetamine use (overall 12%) was present almost entirely in Cluster 1, with a high representation of connected users (alters for using egos mean 0.4, SD 0.7). Methamphetamine use was completely absent from Cluster 2. In the less dense part of Cluster 1, methamphetamine users also used heroin (Cluster 1 19%), forming chains of users. Powder cocaine users (overall 17%) were dispersed more widely across the network and were infrequently connected with each other. Crack cocaine (overall 43%) was common in Cluster 2 (70%), where almost all users were connected (alters for using egos mean 0.9, SD 0.8). It was noticeably absent from the less dense part of Cluster 1, i.e. among the methamphetamine and heroin users making up this part of the cluster. Cannabis use was also common (overall 29%), mainly found in the dense part of Cluster 1 (46%), generally with 2–3 connected users. Alcohol

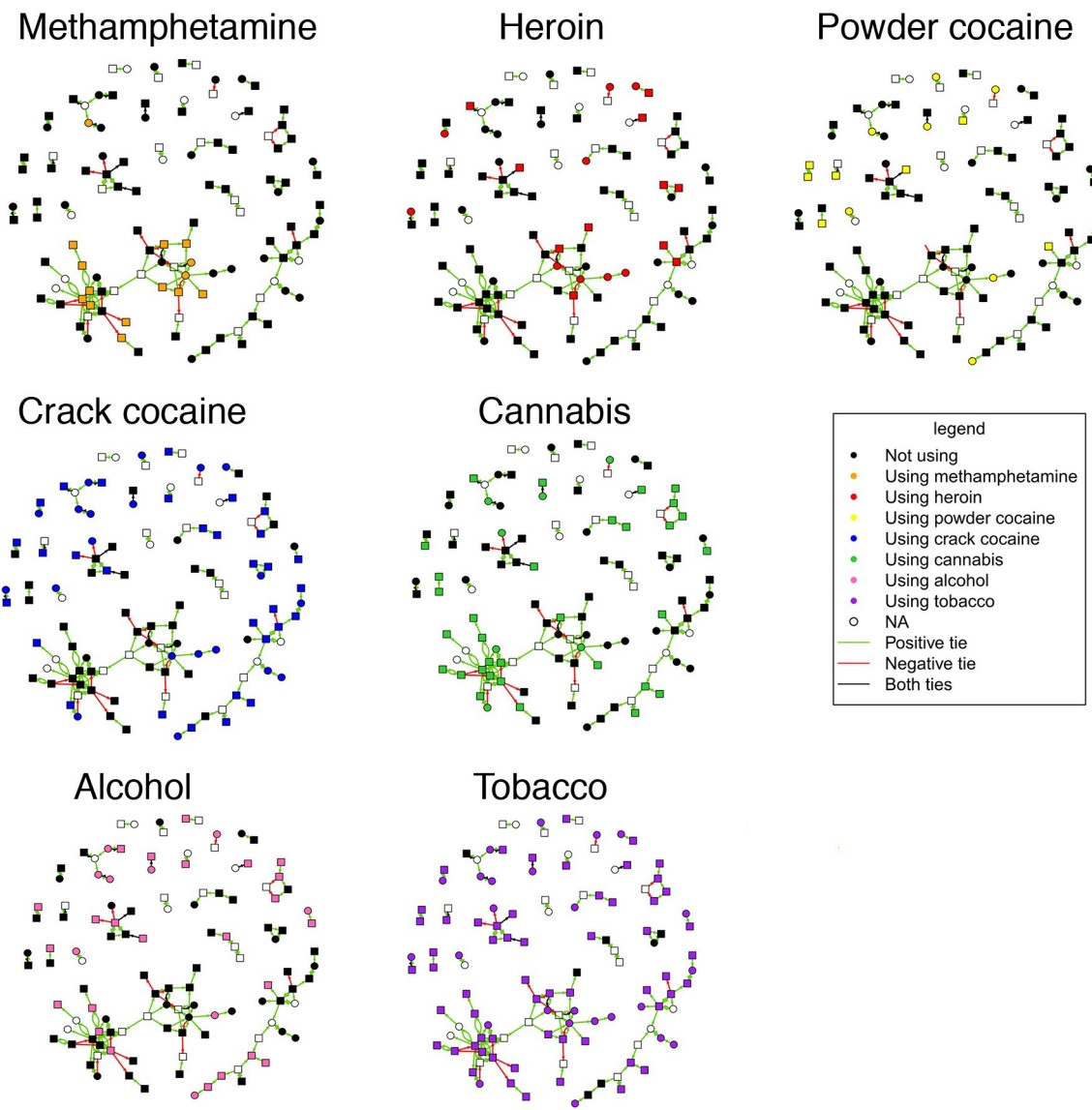

**Fig 2. Network diagram (n = 118) of ego substance use in the first month.** Graph layout is equivalent to Fig 1. Node colour indicates substance use type.

was less common, but concentrated in the dense parts of Cluster 1 (14%). Tobacco was highly prevalent in all parts of the network (overall 69%).

### Association of ego and alter substance use

Table 3 summarizes the results of the additive logistic mixed effects models for ego substance use over time. Since the unadjusted model yielded similar results, only adjusted models including month and ego substance use appear. Overall, ego substance use did not change significantly over the six-month period. Ego heroin use was significantly predicted by alter heroin use. Similar relationships were observed for cannabis use as well as crack cocaine use. In contrast, ego methamphetamine, powder cocaine, and alcohol use were not associated with alter use. These estimates were similar when limited to a subset of participants injecting methamphetamine, heroin or powder cocaine (S3 and S4 Tables).

**Table 2. Substance use in the first month for egos, for all using alters regardless of ego use, and for all using alters for an ego using the same substance.**

|  | MA | Heroin | Cocaine powder | Cocaine crack | Cannabis | Alcohol | Tobacco |
|---|---|---|---|---|---|---|---|
| **Full network** (n = 201) |  |  |  |  |  |  |  |
| Ego (%) user / nonuser / NA | 12 / 62 / 25 | 18 / 58 / 24 | 17 / 59 / 24 | 43 / 33 / 24 | 29 / 47 / 24 | 26 / 50 / 24 | 69 / 7 / 21 |
| Alter mean (SD) | 0.17 (0.62) | 0.12 (0.40) | 0.07 (0.26) | 0.28 (0.56) | 0.28 (0.80) | 0.20 (0.47) | 0.54 (1.07) |
| Alter with ego user mean (SD) | 0.07 (0.35) | 0.05 (0.26) | 0.01 (0.10) | 0.14 (0.38) | 0.17 (0.60) | 0.07 (0.25) | 0.41 (0.94) |
| **Cluster 1** (n = 37) |  |  |  |  |  |  |  |
| Ego (%) user, nonuser, NA | 38 / 43 / 19 | 19 / 62 / 19 | 3 / 78 / 19 | 19 / 62 / 19 | 46 / 35 / 19 | 14 / 68 / 19 | 78 / 3 / 19 |
| Alter Mean (SD) | 0.86 (1.21) | 0.38 (0.72) | 0.03 (0.16) | 0.24 (0.43) | 1.05 (1.47) | 0.35 (0.68) | 1.54 (1.76) |
| Alter with ego user mean (SD) | 0.38 (0.71) | 0.19 (0.41) | 0.00 (0.00) | 0.11 (0.22) | 0.73 (1.08) | 0.11 (0.28) | 1.19 (1.54) |
| **Cluster 2** (n = 20) |  |  |  |  |  |  |  |
| Ego (%) User / nonuser / NA | 0 / 80 / 20 | 10 / 70 / 20 | 15 / 65 / 20 | 70 / 10 / 20 | 20 / 60 / 20 | 30 / 50 / 20 | 70 / 10 / 20 |
| Alter Mean (SD) | 0.00 | 0.15 (0.37) | 0.20 (0.41) | 0.85 (0.81) | 0.25 (0.55) | 0.35 (0.59) | 0.85 (0.93) |
| Alter with ego user mean (SD) | 0.00 (0.00) | 0.05 (0.16) | 0.00 (0.00) | 0.55 (0.74) | 0.05 (0.20) | 0.15 (0.28) | 0.55 (0.87) |
| **Small components** (n = 61) |  |  |  |  |  |  |  |
| Ego (%) user / nonuser / NA | 2 / 70 / 28 | 18 / 56 / 26 | 18 / 56 / 26 | 49 / 25 / 26 | 30 / 44 / 26 | 36 / 38 / 26 | 67 / 7 / 26 |
| Alter mean (SD) | 0.03 (0.18) | 0.13 (0.34) | 0.15 (0.36) | 0.49 (0.67) | 0.21 (0.49) | 0.33 (0.51) | 0.57 (0.81) |
| Alter with ego user mean (SD) | 0.00 (0.00) | 0.03 (0.17) | 0.03 (0.17) | 0.21 (0.40) | 0.10 (0.26) | 0.13 (0.29) | 0.44 (0.72) |
| **Isolates** (n = 83) |  |  |  |  |  |  |  |
| Ego (%) user / nonuser / NA | 12 / 60 / 28 | 20 / 54 / 25 | 23 / 52 / 25 | 42 / 33 / 25 | 24 / 51 / 25 | 24 / 51 / 25 | 66 / 8 / 25 |

MA: methamphetamine, NA: not available

For alters using multiple substances concurrently (Table 4), alter cannabis use was associated with lower ego heroin use. Alter cannabis use was also associated with lower ego crack cocaine use, and vice versa.

**Table 3. Logistic mixed effects modelling results for ego substance use association with alter use of the same substance over six months (n = 118).** Adjusted models include the two predictors: month, and alter substance use, in one model.

| Factor | Odds ratio | 95% confidence interval | p-value |
|---|---|---|---|
| Alter methamphetamine use | 0.23 | 0.02–2.24 | 0.29 |
| Month | 1.24 | 0.89–1.73 | 0.28 |
| Alter heroin use | 19.06 | 4.62–78.58 | <0.0001 |
| Month | 0.97 | 0.79–1.20 | 0.83 |
| Alter cannabis use | 4.79 | 2.45–9.38 | <0.0001 |
| Month | 0.81 | 0.69–0.95 | 0.03 |
| Alter powder cocaine use | 0.06 | 0.01–1.08 | 0.11 |
| Month | 1.15 | 0.86–1.54 | 0.44 |
| Alter crack cocaine use | 5.67 | 1.56–20.59 | 0.03 |
| Month | 0.85 | 0.68–1.05 | 0.21 |
| Alter alcohol use | 1.56 | 0.86–2.84 | 0.22 |
| Month | 0.93 | 0.81–1.07 | 0.39 |

**Table 4. Additive logistic mixed effects modelling results with three predictors for ego substance use (cannabis, crack cocaine) in month 1 (October 2010) (n = 118).**
The unadjusted models account for the predictors month, alter same substance use and alter different substance use separately whereas the adjusted models combine these three predictors in one model.

| Factor | Unadjusted Models | | | Adjusted Models | | |
|---|---|---|---|---|---|---|
| | OR | 95% CI | p-value | OR | 95% CI | p-value |
| Ego cannabis use | | | | | | |
| Alter cannabis use | 4.93 | 2.54–9.56 | <0.001 | 5.24 | 2.66–10.35 | <0.0001 |
| Month | 0.78 | 0.67–0.92 | 0.01 | 0.80 | 0.69–0.94 | 0.02 |
| Alter heroin use | 0.49 | 0.20–1.19 | 0.18 | 0.35 | 0.15–0.83 | <0.05 |
| Ego cannabis use | | | | | | |
| Alter cannabis use | 4.93 | 2.54–9.56 | <0.001 | 6.13 | 3.11–12.05 | <0.0001 |
| Month | 0.78 | 0.67–0.92 | 0.01 | 0.80 | 0.69–0.94 | 0.02 |
| Alter crack cocaine use | 0.43 | 0.20–0.96 | 0.09 | 0.25 | 0.11–0.55 | 0.004 |
| Ego crack cocaine use | | | | | | |
| Alter crack cocaine use | 5.83 | 1.71–19.81 | 0.02 | 10.54 | 2.69–41.24 | 0.005 |
| Month | 0.83 | 0.67–1.02 | 0.14 | 0.81 | 0.64–1.01 | 0.11 |
| Alter cannabis use | 0.19 | 0.07–0.50 | 0.005 | 0.10 | 0.03–0.31 | 0.0001 |

Permutation analysis also provided evidence that dispersion of substance use, and the association between ego and alter use was not random for all substances. Neither the dispersion of methamphetamine (ego for using alters p<0.001) nor cannabis (ego for using alters p<0.001) users was random. Since the sample size is relatively small and resulting null distributions may underestimate the size of the tail especially if a hidden structure in the underlying data is neglected, these results need to be considered tentatively. The distributions including p-values for all egos can be found in figures S1 and S2 Figs. Results for using egos only are displayed in figures S3 and S4 Figs.

## Discussion

This observational study examined the dispersion of use of different substances among tenant social support networks formed by people living in precarious housing. Social support networks largely corresponded to the place of residence, and were characterized by demographic homogeneity. Dispersion of substance use related to the network topology, and differed according to substance used. Personal use of heroin, crack cocaine, and cannabis, but not powder cocaine, tobacco or alcohol, were associated with use of that substance by alters. Alter and ego methamphetamine use was highly associated, but only in a segment of the social network where use was densely concentrated.

The observed network topology included clusters linked with residences, and with greater demographic similarity within the cluster. Spatial proximity and similarities in age and gender may give rise to shared substance use patterns. However, the pre-existing use of substances, and the 6-month duration of observation are features that prevent disentangling risk factors. Longer periods of observation may be required to demonstrate a role for spatial proximity, as reported on a greater scale for methamphetamine use on California [22]. Shorter periods such as used here may be sufficient to detect more dramatic consequences such as the introduction of crack cocaine in New York City in the mid-1980s, or following the completion of the present observation period, the introduction of fentanyl in the neighborhood studied here [51,52]. In general, our findings support the role for network topology in the dispersion of potentially harmful behaviors [30].

Personal and alter substance use were associated for heroin, cannabis and crack cocaine. The observation that personal substance use is affected by the use of the peers is well established [9,11,53,54]. The present observations suggest that this association is dependent on the underlying topology of the network. Full network analyses for methamphetamine did not demonstrate an association with alter substance use. However, a closer look at the network topology revealed that methamphetamine users were almost entirely in Cluster 1, and were indeed highly interconnected. While methamphetamine use may generally not be primarily dependent on alters, they can be relevant given a dense underlying topology. In contrast, for crack cocaine, there was overall evidence for the relevance of alters; yet little support for this finding appeared in Cluster 1. Crack cocaine was the most commonly used substance in Cluster 2—among the older and more demographically homogeneous residents of Hotel B, suggesting demographic features as well as proximity may play important roles. In other studies, the latter has been found to have a negative impact on ongoing substance use since support was likely to take the form of assisting with substance procurement and less likely to involve encouragement for abstinence [54]. Here, network topology was tied to spatial proximity. At the same time, some personal substance use was associated with alters but only in the context of the network topology. These findings highlight the complex interplay of individual as well as social factors in substance use in socially marginalized people living in precarious housing.

Concurrent use of substances creates complex pharmacologies that may be of interest as clues to developing substitution therapies, or novel treatments [55]. Beyond the associations between egos and alters for use of the same substance, two reciprocal associations were observed. Cannabis use in alters was associated with lower use of crack cocaine, and lower use of opioids in egos. Steps towards a less harmful ecology of substance use, considering the full range of substances used, the relationships to an individual's support network, and the topology of the network itself may be needed for the vexing problems of tri-morbidity in socially marginalized people.

Limitations of the study include sample size and assessments, of substance use, and of network variables. The sample size is similar to that in other studies, and has the benefit of repeated assessment over time. However, for pragmatic reasons the network was assumed to be fixed over the six-month observation period. Self-report was used to obtain substance use information; although not used for analytic purposes, urine drug screen showed a high concordance with self-report. As with any network study, the network structure may be subject to more changes than were feasible to capture [32]. We limited the focus to study participants. Lack of information on supporters from outside the study is a limitation. The study findings indicate potential interdependencies between the substances used and the mode of administration, such as injection. These and other factors need to be considered in the design of future research [56,57].

## Conclusions

The findings highlight the complexity of social influences affecting personal substance use. Geographic proximity of demographically similar residents had a crucial impact on the topology of the support network. Substance use of supporters was found to be relevant to personal substance use, depending on the underlying network topology and the substance consumed. The association between social network clusters, substance use behaviors, and hotels suggests that prevention and harm reduction efforts should be targeted at all of the residents of a hotel and reflect the specifics of substance use in that hotel. This underscores the potential value of collecting social network data for designing effective intervention strategies [14,32,56,58]. Understanding the influence of supporters and peer leaders within the social networks could

help in disseminating behaviors to reduce harm. It has been shown that treatment programs for substance use are more successful if the patient's social network can be restructured to minimize the number of substance-using peers [13,20,59]. Combined with the results from this study, this suggests that it would be important to find alternate accommodation for individuals seeking treatment. Removing them from the substance-using environments within the hotels would mitigate the risk for future relapse.

## Supporting information

**S1 Appendix. Custom-made calculations for age and gender disparity.**
(PDF)

**S2 Appendix. Mixed effect modelling and results.**
(PDF)

**S3 Appendix. Adapted permutation analysis.**
(PDF)

**S4 Appendix. References for supplementary material.**
(PDF)

**S1 Fig. Plots of mean alter substance use for all egos where personal substance use was randomised (10,000 iterations, n = 201).** The x- axis represents the density, the y-axis represents the mean for all alters. The dotted line indicates observed mean alter substance use for all egos.
(PDF)

**S2 Fig. Plots of standard deviation of alter substance use for all egos where personal substance use was randomised (10,000 iterations, n = 201).** The x-axis represents the density, the y-axis represents the standard deviation for all alters. The dotted line indicates observed standard deviation of alter substance use for all egos.
(PDF)

**S3 Fig. Plots of mean of alter substance use for egos using the same substance where personal substance use was randomised (10,000 iterations, n = 201).** The x-axis represents the density, the y-axis represents the mean for all alters. The dotted line indicates observed mean alter substance use for egos using the same substance.
(PDF)

**S4 Fig. Plots of standard deviation of alter substance use for egos using the same substance where personal substance use was randomised (10,000 iterations, n = 201).** The x-axis represents the density, the y-axis represents the standard deviation for all alters. The dotted line indicates observed standard deviation of alter substance use for egos using the same substance.
(PDF)

**S1 Table. Demographics and network characteristics of four different graph components.**
(PDF)

**S2 Table. Substance use in the first month for egos, for all using alters regardless of ego use, and for all using alters for an ego using the same substance.**
(PDF)

**S3 Table. Ego and alter same substance injection use.** Additive mixed effects modelling results for ego injection substance use (methamphetamine, heroin, powdered cocaine) in month 1 (October 2010) (n = 118). The unadjusted models account for the predictors month i.e. the time of six months and alter injection substance use separately whereas the adjusted

models combine these two predictors in one model.
(PDF)

**S4 Table. Ego and alter injection use.** Additive logistic mixed effects modelling results with three predictors for ego injection substance use (heroin) in month 1 (October 2010) (n = 118). The unadjusted models account for the predictors month i.e. the time of six months, alter injection substance use and another alter injection substance use separately whereas the adjusted models combine these three predictors in one model.
(PDF)

## Author Contributions

**Conceptualization:** Andrea A. Jones, Sam Seyedin, Christopher Siu, Alasdair M. Barr, William J. Panenka, Allen E. Thornton, William G. Honer, Alexander R. Rutherford.

**Data curation:** Andrea A. Jones, Sam Seyedin, Christopher Siu.

**Formal analysis:** Verena Knerich.

**Funding acquisition:** Alasdair M. Barr, William J. Panenka, Allen E. Thornton, William G. Honer.

**Methodology:** Verena Knerich, Andrea A. Jones.

**Software:** Louie Dinh, Sara Mostafavi.

**Supervision:** Andrea A. Jones, William G. Honer, Alexander R. Rutherford.

**Writing – original draft:** Verena Knerich.

**Writing – review & editing:** Andrea A. Jones, William G. Honer, Alexander R. Rutherford.

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
