## [Decision Letter · Decision Letter 0]

18 Jul 2019

PONE-D-19-16818

Social and structural factors associated with substance use within the support network of adults living in precarious housing in a socially marginalized neighborhood of Vancouver, Canada

PLOS ONE

Dear Dr. Rutherford,

Thank you for submitting your manuscript to PLOS ONE. After careful consideration, we feel that it has merit but does not fully meet PLOS ONE’s publication criteria as it currently stands. Therefore, we invite you to submit a revised version of the manuscript that addresses the points raised during the review process.

ACADEMIC EDITOR: Please insert comments here and delete this placeholder text when finished. Be sure to:

Both reviewers found this to be a well-written and informative study.  However, Reviewer 1 recommended expansion o the introduction if this is to be accepted as a regular research article, and Reviewer 2 recommended expansion of the discussion of implications for interventions.  The editor is in agreement with these two recommendations. 

We would appreciate receiving your revised manuscript by Sep 01 2019 11:59PM. To enhance the reproducibility of your results, we recommend that if applicable you deposit your laboratory protocols in protocols.io, where a protocol can be assigned its own identifier (DOI) such that it can be cited independently in the future. For instructions see: http://journals.plos.org/plosone/s/submission-guidelines#loc-laboratory-protocols

We look forward to receiving your revised manuscript.

Kind regards,

Lawrence Palinkas

Academic Editor

PLOS ONE

2. Thank you for stating the following in the Competing Interests section: "I have read the journal's policy and the authors of this manuscript have the following competing interests:

AMB has received consulting fees or sat on advisory boards for Bristol-Myers Squibb, Eli Lilly, and Roche.

WJP sat on paid advisory boards for Vitality Biopharma, Medipure Pharmaceuticals, and Vinergy Resources; has sat on the board of directors of Abbatis Bioceuticals; and is owner of Translational Life Sciences.

WGH received consulting fees or sat on paid advisory boards for: the Canadian Agency for Drugs and Technology in Health, AlphaSights, Guidepoint, In Silico, Translational Life Sciences, Otsuka, Lundbeck, and Newron."

Reviewers' comments:

Reviewer's Responses to Questions

**Comments to the Author**

1. Is the manuscript technically sound, and do the data support the conclusions?

Reviewer #1: Yes

Reviewer #2: Yes

2. Has the statistical analysis been performed appropriately and rigorously? 

Reviewer #1: Yes

Reviewer #2: Yes

3. Have the authors made all data underlying the findings in their manuscript fully available?

Reviewer #1: No

Reviewer #2: Yes

4. Is the manuscript presented in an intelligible fashion and written in standard English?

Reviewer #1: Yes

Reviewer #2: Yes

5. Review Comments to the Author

Reviewer #1: In general this is a well-written and clear paper with interesting and meaningful results. In particular the authors do an excellent job of describing their methods and results in a way that will be understood by a broad audience. Any suggestions for edits are only relevant if this paper is not meant to be published as a brief report.

If this paper is not a brief report, then the introduction needs to be expanded to further explore how existing literature frames the current study. This should include literature that is specific to substance use in networks, homelessness/housing, and the assortativity characteristics you assess. If the introduction is expanded, it would also be helpful to articulate a set of research questions that clearly lead to your analytic plan.

Methods are extensive, appropriate for the study, and clearly described. The authors do a nice job outlining rigorous methods in a way that will be understandable to a broad audience.

The discussion is concise, but provides a nice, comprehensible overview of findings and implications. If the paper is expanded, I would like to see more specific recommendations for intervention, housing, and/or practice from the perspective of the authors.

Reviewer #2: This paper provides a valuable exploration into how the structure of networks and peer relationships impact the substance using behavior of precariously housed individuals recruited single room occupancy hotels in Vancouver. Overall the paper is quite excellent. The methods are well described and the social network analysis is appropriate to the questions raised by the authors.

My only minor criticism is with the discussion section. The authors make a point in both the abstract and the introduction of framing the importance of this work around the potential to inform interventions. Yet, the discussion of intervention direction is very superficial. Essentially, they say in one sentence on p. 20 line 370-371 “Concurrent use of substances creates complex phamacologies that may be of interest as clues to developing substitution therapies, or novel treatments.” This is really insufficient given the frame of the paper. I would recommend several citations below by Rice and his colleagues which explore similar network issues among homeless youth and provide more rigorous direction for interventions.

Rice, E., & Rhoades, H. (2013). How should network-based prevention for homeless youth be implemented?. Addiction (Abingdon, England), 108(9), 1625.

Rice, E., Barman-Adhikari, A., Milburn, N. G., & Monro, W. (2012). Position-specific HIV risk in a large network of homeless youths. American journal of public health, 102(1), 141-147.

Barman-Adhikari, A., Rice, E., Winetrobe, H., & Petering, R. (2015). Social network correlates of methamphetamine, heroin, and cocaine use in a sociometric network of homeless youth. Journal of the Society for Social Work and Research, 6(3), 433-457.

Rhoades, H., La Motte-Kerr, W., Duan, L., Woo, D., Rice, E., Henwood, B., ... & Wenzel, S. L. (2018). Social networks and substance use after transitioning into permanentsupportive housing. Drug and alcohol dependence, 191, 63-69.

6. PLOS authors have the option to publish the peer review history of their article (what does this mean?). If published, this will include your full peer review and any attached files.

Reviewer #1: Yes: Harmony R Rhoades

Reviewer #2: Yes: Eric Rice

---

## [Author Response · Author response to Decision Letter 0]

1 Sep 2019

Response to Reviewer 1:

The introduction has been expanded and a number of references added to better frame the study in the context of previous work on substance use and social network analysis. We have also reviewed previous work on using social network analysis to investigate prevention strategies and improve treatment programs. A conclusion has been added to summarize the implications of our work for prevention, harm reduction, and treatment.

Response to Reviewer 2:

As mentioned above, a conclusion has been added to expand on the implications of our work for using social network analysis to improve prevention, harm reduction, and treatment. Furthermore, the references that were suggested have been added, as well as a number of additional references. We thank the reviewer for drawing this body of work to our attention.

---

## [Editor Report · Decision Letter 1]

4 Sep 2019

Social and structural factors associated with substance use within the support network of adults living in precarious housing in a socially marginalized neighborhood of Vancouver, Canada

PONE-D-19-16818R1

Dear Dr. Rutherford,

We are pleased to inform you that your manuscript has been judged scientifically suitable for publication and will be formally accepted for publication once it complies with all outstanding technical requirements.

With kind regards,

Lawrence Palinkas

Academic Editor

PLOS ONE

Additional Editor Comments (optional):

The revised manuscript is responsive to the suggestions provided by the viewers. This should make for a fine contribution to the literature.
---

## [Editor Report · Acceptance letter]

9 Sep 2019

PONE-D-19-16818R1 

Social and structural factors associated with substance use within the support network of adults living in precarious housing in a socially marginalized neighborhood of Vancouver, Canada 

Dear Dr. Rutherford:

I am pleased to inform you that your manuscript has been deemed suitable for publication in PLOS ONE. Congratulations! Your manuscript is now with our production department. 

With kind regards,

on behalf of

Dr. Lawrence Palinkas 

Academic Editor

PLOS ONE